# Implementation of a Recycled Polypropylene Homopolymer Material for Use in Additive Manufacturing

**Jozef Dobránsky** [1,*] , **Martin Pollák** [1] , **Luboš Běhálek** [2] and **Jozef Svetlík** [3]

1   Faculty of Manufacturing Technologies with a Seat in Presov, Technical University of Kosice, 080 01 Presov, Slovakia; martin.pollak@tuke.sk
2   Faculty of Mechanical Engineering, Technical University of Liberec, 460 01 Liberec, Czech Republic; lubos.behalek@tul.cz
3   Faculty of Mechanical Engineering, Technical University of Kosice, 040 01 Kosice, Slovakia; jozef.svetlik@tuke.sk
*   Correspondence: jozef.dobransky@tuke.sk; Tel.: +421-55-602-6350

**Abstract:** The main objective of the presented scientific article is to define the mechanical properties of polypropylene homopolymer with a prescribed percentage ratio of recycled granulate. The chosen material is intended for injection molding and especially for the production of products made by additive technologies. Experimental verification of the mechanical properties was realized by testing samples produced with various concentrations of the recycled material. Experimental samples underwent tests to obtain the mechanical properties of the produced new material. These tests included rheological tests, tensile and flexural tests as well as and Charpy impact toughness tests. These mechanical tests were conducted according to ISO standards valid for the individual testing method. Testing methods were carried out using prescribed numbers of testing samples. The presented scientific article is also focused on changes in microstructures of testing materials in relation to the percentage ratio of recycled granulate. Recycled granulate of thermoplastic was not necessity for additional modifications.

**Keywords:** additive technologies; granulate; material properties; polymer; recycling

## 1. Introduction

Thermoplastic injection technology is currently among the most commonly used plastics processing technologies, including for the production of plastic products by additive technology. Thanks to the wide range of uses of thermoplastics, especially in the automotive and electronics industries, the technology is ever advancing. There are many factors affecting the final quality of plastic products. The most significant factors affecting the quality of final products include technological parameters of injection molding machines, which are directly related to the production process. The production process of injection molding is affected by a number of parameters, which are interrelated and interdependent [1,2].

Among the biggest disadvantage of plastics is their long life, which has a negative impact on the environment. There have been efforts by manufacturers to implement into the production of plastics recycled or of regranulated materials. Evaluating the quality of products made of materials containing regranulate is possible only by experimental research [3].

Plastics can be defined as macromolecular substances that can be shaped by heat or pressure, or by both agents simultaneously [4,5].

The various types of plastics have their distinctive functional and processing properties. They may be partially varied or adjusted using of additives. Their properties are mainly evaluated from a functional perspective [6,7], including the following:

- mechanical strength for long-term or short-term static and dynamic loads,

- electrical properties such as dielectric strength, conductivity, etc.,
- chemical resistance to various chemical agents, for the food industry, and
- optical properties such as transparency, color, gloss, etc.

The processing aspect is equally important. Significant properties include

- the fluidity, which affects critical wall thickness of a product, concept of the molding, and size of the inlet and also affects tempering of the mold (optimum temperature of the tool in relation to the processing of plastics as well as construction and technological parameters),
- shrink size, which determines the accuracy of product manufacturing, and
- sensitivity to technological parameters of manufacturing equipment.

For enhancement of polymeric materials processing, the most common additives are added to the polymer.

Injection molding is a cyclic process, with an average cycle time from 15 to 120 s for thermoplastics. The cycle time depends on the size and properties of the polymer product. Product weight ranges from a few grams to 25 kg.

The process of injection is as follows: granular plastics are supplied to a hopper with injection machine working parts (piston, screw), which transports the material into the melting chamber where, as a result of the simultaneous actions of friction and temperature, the plastic melts. The melted mass is injected into the mold cavity, which completely fills and assumes the shape. A compression stage follows to reduce the shrinkage and dimensional changes. The plastic transmits heat to the mold, and cooling solidifies it into the final product. In the final stage, the mold is opened, the product is ejected, and the cycle is repeated [8].

At present, the recycling of various types of materials for their further use and improvement of mechanical and thermal properties is becoming a very important area of research. In their study, Momanyi et al. [9] dealt with the thermomechanical properties of recycled polypropylene and recycled polystyrene. These materials were compared to virgin polypropylene and newly made 1540 polystyrene and 7240 high strength polystyrene. The aim of the study was to determine the hardness and toughness of recycled polymers. Furthermore, the tensile strength, the Young's modulus for their industrial use, and the increase in the recycling rate were measured, leading to a reduction in environmental pollution. The study by Ciro et al. [10] deals with the effect of recycled rubber (RR) in the form of granules with recycled polypropylene/polyethylene (RPP). The properties of these formed mixtures were evaluated by rheometric, thermogravimetric (TGA), scanning electronic microscopy (SEM), differential scanning calorimetric (DSC), melt flow index, and density and tensile analysis. In another article, Maddah [11] focused on the evaluation of polypropylene material (PP) from several perspectives. He focused on this material in order to point out the suitability of its use (advantages and disadvantages), with a focus on its physical and chemical properties. By comparison, he found that polypropylene is the lightest type of plastic with a density of 0.90 g/cm$^3$. Branching into linear polypropylene produces plastics with high modulus, tensile strength, and rigidity as well as excellent heat resistance. Furthermore, electrically conductive PP and PE have excellent electrical and mechanical properties. As in the previous article, Ares et al. [12] investigated the effect of recycled polypropylene (PP) on the rheological, mechanical, and thermal properties of polypropylene wood flour composites at the melting stage. The LVE behavior of polypropylene composites was analyzed using modulus and viscosity curves as well as by studying the thermal and mechanical properties of samples with 10% filler. It was concluded that the added recycled (PP) content did not result in changes in its physical properties or rheology. It can be argued from the results that the new materials can be used to create products that are produced by injection molding (PP) technology and that have significant thermal and mechanical properties.

At present, various types of recycled waste are widely used as raw materials for bitumen treatment. These recycled materials mainly include recycled polypropylene (PP), recycled polystyrene (PS), and recycled low density polyethylene (LDPE). Authors Akkouri

et al. [13] addressed the different thermal and mechanical properties of recycled modified bitumen plastic waste (RPMB) compared to the properties of recycled pure modified bitumen plastic, which were significantly lower in the latter case of recycled plastic. The main goal of the authors' research was to analyze these properties and improve them. Measurements were performed with the use of a commercial styrene-butadiene-styrene elastomer (SBS).

A very important topic of scientific research today is also the use of wood fibers in applications as a filler for thermoplastic polymers. Mazzanti et al. dealt with this topic in their paper [14], where they focused on research of 50 wt. % PP-based WPC toughened with a polypropylene-based thermoplastic vulcanizate (TPV). To the resulting compound was added 2% TPV to reduce its fragility, and the conditions of its processing were subsequently evaluated. The results were processed and verified by dynamic tensile impact tests. TPV showed almost double the impact strength compared to untreated samples. The authors found that the strength of composites depends on the feeding condition, and conversely the tensile stiffness is relatively unaffected by processing conditions. The result of the investigation was that a minimum of 20 wt. % TPV is required to provide a significant increase in composite toughness and ductility, provided that the material is processed in a starved or flooded condition with a low screw speed. The study by Clemons [15] deals with the formation of matrices of wood-composite materials WPC using a mixture of PE polyurethane and PP polypropylene. For their formation, tests of mechanical properties and morphology of wood-filled composites using several types of elastomers and bonding agents were created. As a result, it was found that unmixed HDPE and PP plastics had different mechanical properties. Polypropylene had a higher modulus of strength and lower impact strength and strains at yielding and failure.

The proposed printhead in the application of the utility model of the Slovak Republic [16] opens up new possibilities in the field of additive technologies. It enables spatial printing of bodies from different types of materials as well as mutual mixing of materials and additives, with everything depending on the required final properties of the printed object. The utility model describes the design solution of a printhead with an extruder for dosing thermoplastic material enabling additive production of bodies from various types of materials and added fillers as well as recycled material. The solution of the printhead with the extruder made in this way can be mounted on the robot arm to ensure additive production using plastic or recycled material in the form of granules, e.g., recycled PET bottles. Similarly, the paper [17] describes the design of a 3D printhead solution with an extruder and its attachment to a robot arm for the implementation of additive production of components of larger dimensions. The article contains a description of individual parts of the design solution of the end head, and on the created simulation, in the environment of the RoboDK software, it points out the possibilities of the implementation of additive production by the proposed solution.

## 2. Description of the Experiment

Experimental samples were produced in the Department of Engineering Technology, Faculty of Mechanical Engineering of the Technical University in Liberec. The selected experimental material was Mosten GB 005 (polypropylene homopolymer). Mosten GB 005 is a polypropylene produced by Unipetrol RPA using INNOVENETM PP gas-phase technology. This material is used for applications such as formed food packages, technical parts, household articles, monoaxially oriented tapes, strings, and staple fiber. Experimental samples from this material were produced on injection molding machine ENGEL Victory 80/25.

Technological parameters were preset according to the material data sheets of selected material. Material properties are shown in Table 1.

**Table 1.** Material properties of Mosten GB 005.

| Physical | Nominal Value | Unit | Test Method |
|---|---|---|---|
| Melt mass-flow rate (MFR) (230 °C/2.16 kg) | 5 | (g/10 min) | ISO 1133 |
| **Hardness** | **Nominal Value** | **Unit** | **Test Method** |
| Shore Hardness (Shore D) | 65 | | ISO 868 |
| **Mechanical** | **Nominal Value** | **Unit** | **Test Method** |
| Tensile Modulus | 1500 | MPa | ISO 527-2 |
| Tensile Stress (Yield) | 34 | MPa | ISO 527-2 |
| Tensile Strain | | | ISO 527-2 |
| Yield | 9 | % | |
| Break | 250 | % | |
| Tensile Creep Modulus | | | ISO 899-1 |
| 1 h | 1000 | MPa | |
| 1000 h | 400 | MPa | |
| Flexural modulus | 1600 | MPa | ISO 178 |
| **Impact** | **Nominal Value** | **Unit** | **Test Method** |
| Charpy Notched Impact Strength (23 °C) | 4 | kJ.m$^{-2}$ | ISO 179/1 |
| **Thermal** | **Nominal Value** | **Unit** | **Test Method** |
| Heat Deflection Temperature (1.8 MPa) | 55 | °C | ISO 75-2/A |
| Vicat Softening Temperature | 155 | °C | ISO 306 |
| Melting Temperature | 168 to 172 | °C | ISO 11357-3 |
| **Injection** | **Nominal Value** | **Unit** | **Test Method** |
| Processing (Melt) Temperature | 200 to 260 | °C | |

Testing samples were divided into 7 batches according to the percentage ratio of recyclate/virgin material, as follows:

1. 0 wt. %—virgin material,
2. 10 wt. % content of recyclate,
3. 20 wt. % content of recyclate,
4. 30 wt. % content of recyclate,
5. 50 wt. % content of recyclate,
6. 70 wt. % content of recyclate, and
7. 100 wt. % recyclate.

The experimental procedure was performed according to appropriate standards for each type of test. The flexural properties test was conducted on 5 testing samples, and remaining tests were conducted on 10 testing samples.

The realized tests were as follows:

- Test of the rheological properties: EN ISO 1133:2006
- Hardness test: EN ISO 868:2003
- Impact strength (notch toughness): EN ISO 179-1/1eU
- Test of flexural properties: EN ISO 178:203
- Test of thermal properties: EN ISO 11357-1:2010

Rheological properties testing was performed according to EN ISO 1133:2006 using rheometer CEAST. Experimental samples with a percentage ratio from 0% to 100% as mentioned above were tested for melt mass-volume rate (MVR) cm$^3$/10 min, which describes the flow properties of plastic materials. In MVR testing plastic material is pressed through a capillary with a diameter of 2095 mm and an 8 mm length.

For each class, 10 measurements were performed, and average values were subsequently calculated and graphical dependence was established.

Experimental conditions for Mosten GB 005 were as follows:

- Melting chamber temperature—230 °C
- Nominal load—2.16 kg
- Length of piston—25 mm

## 3. Results and Discussion

*3.1. Evaluation of the Rheological Properties (MVR—Melt Volume Flow Rate)*

Figure 1 shows the graphical dependence of MVR on the percentage ratio of recyclate material.

$$MVR = -1.7{\cdot}10^{-6} \cdot x^3 - 3.29{\cdot}10^{-5} \cdot x^2 + 0.026 \cdot x + 6.93 \tag{1}$$

where MVR—melt volume flow rate, x—percentage of recycled granulate.

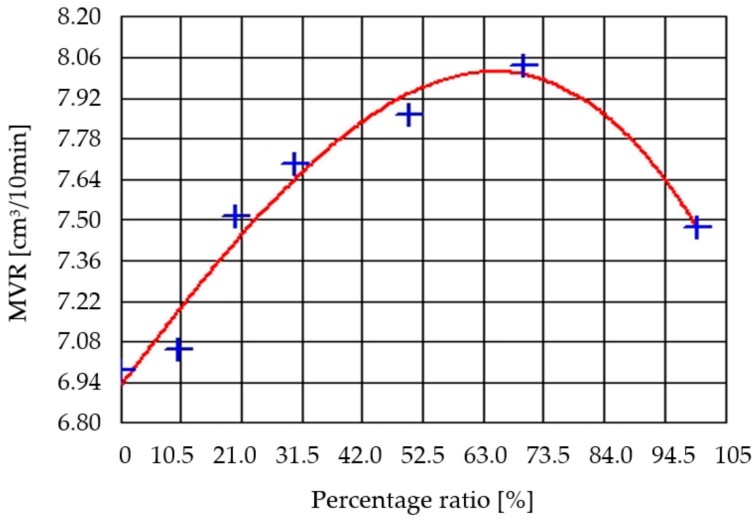

**Figure 1.** Graphical dependence of MVR on percentage ratio of the recyclate.

The correlation index of measured (blue points) and calculated (red curve) values is 97.89%, dispersion is 0.1313, and standard deviation is 0.36. Measured values of MVR are shown in Table 2. All calculation of variations and deviations were realized based on the following assumptions and equations. $X$ is a random variable, which takes the final or countless many values. Then we define dispersion as

$$VAR[X] = \frac{\sum_{i=1}^{n} (x_i - x)^2}{n-1} \tag{2}$$

and deviation as

$$DEV[X] = \sqrt{\frac{\sum_{i=1}^{n} (x_i - x)^2}{n-1}} \tag{3}$$

**Table 2.** Measured values of MVR for Mosten GB 005.

| Sample | Number of Material Batch—Percentage Ratio of Recycled Material | | | | | | |
|---|---|---|---|---|---|---|---|
| | 1—0% | 2—10% | 3—20% | 4—30% | 5—50% | 6—70% | 7—100% |
| 1 | 6.90 | 6.95 | 7.31 | 7.53 | 7.70 | 7.86 | 7.25 |
| 2 | 6.92 | 6.96 | 7.44 | 7.58 | 7.83 | 7.88 | 7.29 |
| 3 | 7.02 | 7.09 | 7.46 | 7.53 | 7.83 | 7.98 | 7.35 |
| 4 | 7.03 | 7.05 | 7.41 | 7.69 | 7.81 | 8.00 | 7.41 |
| 5 | 7.00 | 7.15 | 7.48 | 7.74 | 7.81 | 8.00 | 7.43 |
| 6 | 6.97 | 7.02 | 7.56 | 7.81 | 7.90 | 8.07 | 7.48 |
| 7 | 7.00 | 7.05 | 7.58 | 7.79 | 8.07 | 8.09 | 7.56 |
| 8 | 7.02 | 7.11 | 7.58 | 7.82 | 7.98 | 8.15 | 7.67 |
| 9 | 7.01 | 7.15 | 7.62 | 7.77 | 7.88 | 8.16 | 7.72 |
| 10 | 7.05 | 7.22 | 7.61 | 7.79 | 7.80 | 8.15 | 7.63 |
| **Average [cm$^3$/10 min]** | **6.98** | **7.05** | **7.51** | **7.69** | **7.86** | **8.03** | **7.47** |

Based on the results of MVR as shown in Figure 1, it can be stated that material batch no. 1 achieves the lowest value of MVR, at 6.98 cm$^3$/10 min. Pure material containing no regranulate passes through the capillary at the time with the smallest amount of melt. The addition of 10% regrind into the base material (batch no. 2) results in a slightly increased MVR. For material batch no. 3, MVR increased 0.46 cm$^3$/10 min compared to the material batch no. 2. A gradual increase was recorded for material with regranulate containing 30, 50, and 70% (batch nos. 4, 5, and 6), which is also evident from the graph. Material batch no. 7 (regranulate 100%) showed an average decrease in the values MVR of 0.50 cm$^3$/10 min compared to material batch no. 6. The results also confirm the conclusions of Sahli et al. [18], Dai et al. [19], and Spicker et al. [20].

### 3.2. Evaluation of the Tensile Properties

Tensile property testing was performed using tensile machine Hounsfield H10KT and software QMat according in accordance with standard STN EN ISO 527—1.2. For each batch, 10 measurements were performed, and average values were subsequently calculated and graphical dependence established.

Evaluated tensile properties parameters were as follows:

- Yield stress $\sigma_y$ (MPa)
- Nominal elongation $\varepsilon_t$ (%)
- Nominal elongation at fracture $\varepsilon_{tB}$ (%)
- Tensile strength at fracture $\sigma_B$ (MPa)

Conditions of tensile property testing for Mosten GB 005 were as follows:

- Measured without preloading
- Sensor head 10 kN
- Initial distance between jaws, $L_0$ = 102 mm

Figure 2 shows the graphical dependence of yield stress for various ratios of regranulate material.

$$YP = -32.17 \cdot 10^{-4} \cdot x^2 - 0.04426 \cdot x + 34.34 \tag{4}$$

where YP—yield point, x—percentage of recycled granulate.

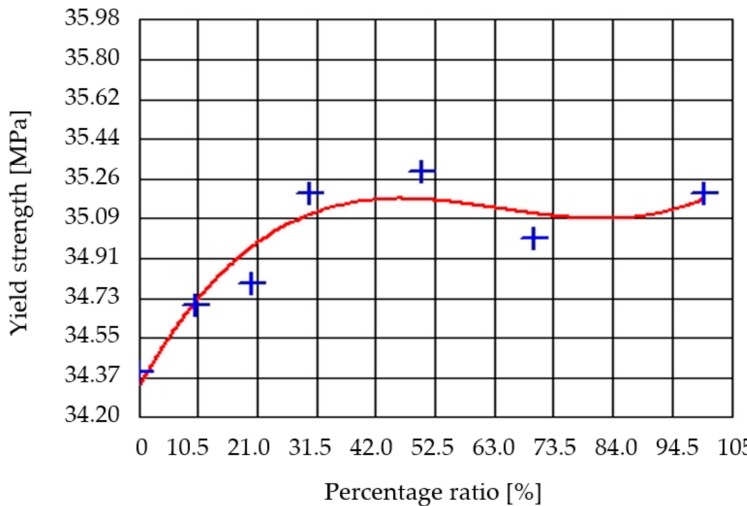

**Figure 2.** Yield strength.

The correlation index of measured (blue points) and calculated (red curve) values is 93.01%, dispersion is 0.091, and standard deviation is 0.306. Measured values of yield stress and average values are shown in Table 3.

**Table 3.** Measured values of $\sigma_y$ for Mosten GB 005.

| Sample | Number of Material Batch—Percentage Ratio of Recycled Material | | | | | | |
|---|---|---|---|---|---|---|---|
| | 1—0% | 2—10% | 3—20% | 4—30% | 5—50% | 6—70% | 7—100% |
| 1 | 35.2 | 35.0 | 34.7 | 35.0 | 35.2 | 35.0 | 35.4 |
| 2 | 34.3 | 35.3 | 34.4 | 34.8 | 35.3 | 35.0 | 35.2 |
| 3 | 34.0 | 34.9 | 34.6 | 35.2 | 35.5 | 35.1 | 35.1 |
| 4 | 34.3 | 34.9 | 34.8 | 35.0 | 35.4 | 35.0 | 35.1 |
| 5 | 34.2 | 34.0 | 34.8 | 35.3 | 35.5 | 35.1 | 35.4 |
| 6 | 34.2 | 34.6 | 34.3 | 35.2 | 35.5 | 35.0 | 35.1 |
| 7 | 34.5 | 34.7 | 35.3 | 35.3 | 35.5 | 35.1 | 35.3 |
| 8 | 34.4 | 34.0 | 34.8 | 35.4 | 35.1 | 35.1 | 35.4 |
| 9 | 34.6 | 34.7 | 35.0 | 35.3 | 35.1 | 35.1 | 35.3 |
| 10 | 34.3 | 34.9 | 35.0 | 35.0 | 34.9 | 35.0 | 34.6 |
| **Average [MPa]** | **34.4** | **34.7** | **34.8** | **35.2** | **35.3** | **35.0** | **35.2** |

Graphical dependence in Figure 2 shows that an increasing ratio of regranulates increases yield strength. For pure material (batch no. 1) containing no regranulate, $\sigma_y$ was 34.4 MPa. Addition of 10% granulate (material batch no. 2) to 50% (material batch no. 5) to the base material resulted in an increase of yield stress from 34.7 MPa to 35.3 MPa, which was the maximum measured value. A slight decrease in yield stress was recorded for material batch no. 6, where the value decreased to 35.0 MPa. Increase of the $\sigma_y$ value was observed for material batch no. 7, with a value of 352 MPa. This behavior was in agreement with data observed by Yan et al. [21] and Kada et al. [22].

Figure 3 represents the dependence of nominal elongation on the ratio of regranulate material.

$$E = -2.7 \cdot 10^{-5} \cdot x^2 + 0.00237 \cdot x + 9.77 \tag{5}$$

where E—elongation, x—percentage of recycled granulate.

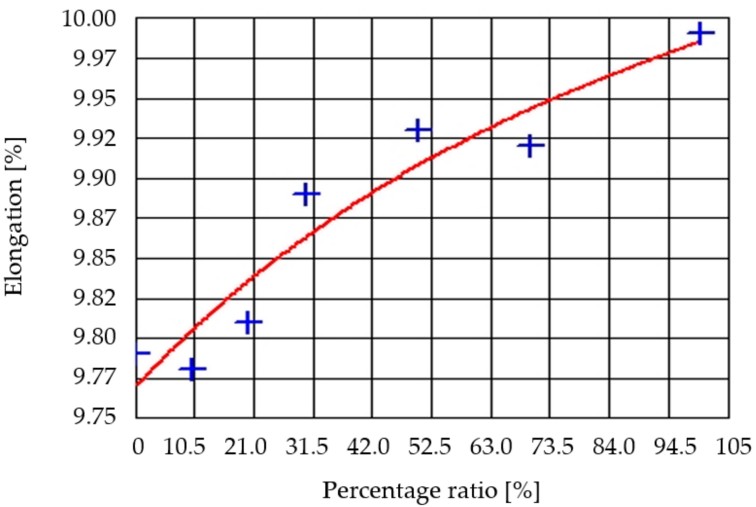

**Figure 3.** Elongation before fraction.

The correlation index of measured (blue points) and calculated (red curve) values is 95.3%, dispersion is 0.00563, and standard deviation is 0.075. Measured and average values of nominal elongation are listed in Table 4.

**Table 4.** Measured values of $\sigma_y$ for Mosten GB 005.

| Sample | Number of Material Batch—Percentage Ratio of Recycled Material | | | | | | |
|---|---|---|---|---|---|---|---|
| | 1—0% | 2—10% | 3—20% | 4—30% | 5—50% | 6—70% | 7—100% |
| 1 | 10.00 | 9.82 | 9.93 | 10.00 | 9.90 | 10.29 | 9.41 |
| 2 | 10.10 | 9.68 | 9.61 | 9.31 | 9.44 | 9.80 | 9.44 |
| 3 | 9.51 | 9.56 | 9.61 | 9.61 | 9.56 | 9.91 | 9.44 |
| 4 | 9.41 | 9.34 | 9.80 | 10.39 | 9.90 | 9.88 | 10.78 |
| 5 | 10.39 | 9.68 | 9.61 | 9.56 | 10.00 | 9.91 | 9.85 |
| 6 | 9.56 | 9.56 | 9.61 | 10.29 | 9.61 | 9.81 | 10.00 |
| 7 | 9.71 | 10.15 | 10.05 | 9.90 | 10.47 | 9.80 | 9.49 |
| 8 | 10.05 | 9.62 | 10.05 | 9.31 | 9.41 | 10.14 | 9.80 |
| 9 | 9.80 | 9.80 | 9.68 | 10.20 | 10.78 | 9.84 | 11.40 |
| 10 | 9.41 | 10.59 | 10.12 | 10.29 | 10.20 | 9.85 | 10.25 |
| **Average [%]** | **9.79** | **9.78** | **9.81** | **9.89** | **9.93** | **9.92** | **9.99** |

The results of the measurements of the nominal elongation varied from 9.78% for the material with 10% content of granulate (batch no. 2) to 9.99% for material with 100% content of granulate (batch no. 7). Material batch no. 1 had a value of 9.79%. A decrease of 0.1% occurred for material batch no. 2 with 10% regrind as compared to the base material. For material batch nos. 3, 4, and 5, there was a gradual increase in values from 9.81% to 9.93%. A further decline of 0.1% was recorded for material batch no. 6 (70% granulate), to 9.92%. An increase was seen for 100% raw plastic (batch no. 7), which had a value of 9.99%. This behavior was in agreement with data observed by Rizvi et al. [23].

Figure 4 shows the graphical dependence of the nominal elongation at fracture on the ratio of regranulate.

$$\text{EF} = 1.17 \cdot 10^{-6} \cdot x^3 + 3.96 \cdot 10^{-4} \cdot x^2 - 0.1 \cdot x + 97.88 \tag{6}$$

where EF—elongation after fracture, x—percentage of recycled granulate.

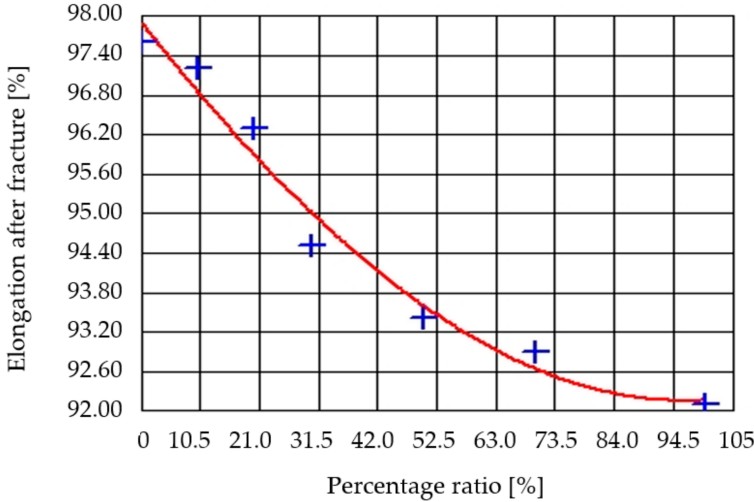

**Figure 4.** Elongation after fracture depending on the percentage ratio of recyclates.

The correlation index of measured (blue points) and calculated (red curve) values is 98.66%, dispersion is 4.11, and standard deviation is 2.028. Measured and average values of nominal elongation at fracture are listed in Table 5.

**Table 5.** Measured values of $\varepsilon_{tB}$ for Mosten GB 005.

| Sample | Number of Material Batch—Percentage Ratio of Recycled Material | | | | | | |
|---|---|---|---|---|---|---|---|
| | **1—0%** | **2—10%** | **3—20%** | **4—30%** | **5—50%** | **6—70%** | **7—100%** |
| 1 | 85.3 | 92.7 | 106.7 | 98.1 | 86.5 | 98.5 | 91.0 |
| 2 | 82.0 | 90.7 | 96.4 | 96.6 | 99.5 | 84.1 | 98.0 |
| 3 | 94.0 | 117.2 | 109.3 | 92.3 | 104.3 | 81.3 | 96.6 |
| 4 | 124.7 | 92.9 | 87.0 | 88.7 | 80.8 | 109.8 | 97.7 |
| 5 | 88.2 | 105.7 | 79.7 | 103.7 | 89.7 | 105.3 | 85.4 |
| 6 | 107.8 | 100.4 | 95.9 | 87.5 | 91.5 | 79.5 | 92.7 |
| 7 | 96.1 | 101.6 | 105.9 | 95.0 | 85.0 | 80.6 | 73.5 |
| 8 | 105.8 | 84.0 | 106.3 | 96.9 | 85.5 | 79.4 | 101.0 |
| 9 | 100.7 | 107.6 | 99.5 | 89.5 | 107.7 | 97.8 | 97.6 |
| 10 | 91.4 | 79.0 | 76.9 | 96.5 | 103.5 | 112.5 | 77.5 |
| **Average [%]** | **97.6** | **97.2** | **96.3** | **94.5** | **93.4** | **92.9** | **92.1** |

The graphical dependence in Figure 4 shows that maximum value of the nominal elongation at fracture 97.6% was reached for material batch no. 1 (pure material). An increase in the ratio of regranulate material caused a decrease to 97.2% in material batch no. 2 (10% regrind) and to the lowest value of 92.1% in material batch no. 7 (100% regranulate). This behavior was in agreement with data observed also by Reixach et al. [24]. The following Figure 5 shows dependence of tensile strength at fracture on the percentage ratio of regranulate.

$$T = -1.2 \cdot 10^{-6} \cdot x^3 + 2.96 \cdot 10^{-4} \cdot x^2 - 0.021 \cdot x + 16.22 \tag{7}$$

where T—tensile strength, x—percentage of recycled granulate.

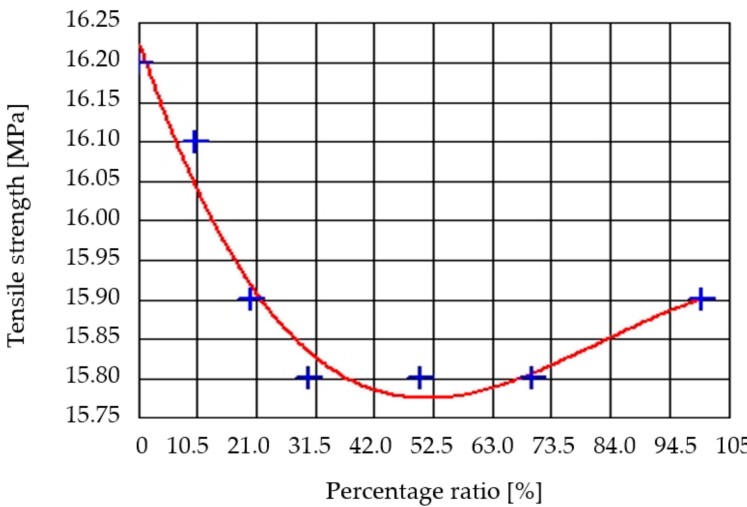

**Figure 5.** Dependence of tensile strength on various percentage ratios of recyclates.

The correlation index of measured (blue points) and calculated (red curve) values is 98.39%, dispersion is 0.022, and standard deviation is 0.149. Measured and average values of tensile strength at fracture are listed in Table 6.

**Table 6.** Measured values of $\sigma_B$ for Mosten GB 005.

| Sample | Number of Material Batch—Percentage Ratio of Recycled Material | | | | | | |
|---|---|---|---|---|---|---|---|
| | 1—0% | 2—10% | 3—20% | 4—30% | 5—50% | 6—70% | 7—100% |
| 1 | 14.60 | 15.70 | 15.11 | 15.04 | 15.04 | 16.35 | 15.04 |
| 2 | 17.66 | 17.10 | 15.70 | 16.31 | 14.95 | 15.49 | 16.88 |
| 3 | 16.95 | 17.10 | 15.98 | 16.61 | 15.64 | 15.75 | 17.63 |
| 4 | 16.39 | 17.02 | 14.81 | 16.49 | 16.35 | 15.30 | 16.16 |
| 5 | 17.29 | 17.48 | 16.05 | 15.45 | 14.96 | 16.17 | 16.65 |
| 6 | 16.88 | 15.02 | 14.96 | 14.82 | 16.01 | 16.84 | 14.40 |
| 7 | 14.45 | 16.76 | 16.99 | 14.73 | 15.26 | 16.59 | 9.60 |
| 8 | 14.14 | 15.49 | 15.90 | 16.05 | 15.11 | 14.69 | 15.30 |
| 9 | 17.14 | 13.95 | 17.10 | 16.80 | 17.17 | 14.66 | 16.91 |
| 10 | 16.73 | 15.11 | 16.46 | 15.98 | 17.59 | 16.73 | 20.17 |
| **Average [MPa]** | **16.2** | **16.1** | **15.9** | **15.8** | **15.8** | **15.8** | **15.9** |

As seen in Figure 5, it can be stated that the maximum value of 16.2 MPa was measured in material batch no. 1 and the minimum of 15.8 MPa was in material batch nos. 4, 5, and 6. A higher ratio of regranulate material resulted in a decrease in the value for a material batch nos. 2 to 5. For materials with 70% and 100% regranulate, the value increased slightly, to 15.8 in material batch no. 6 and to 15.9 in material batch no. 7. The results also confirm the conclusions of Samat et al. [25].

*3.3. Evaluation of the Hardness Properties*

A hardness test was performed using hardness tester Instron 902B with a digital display for hardness readings. The test was performed in accordance with standard EN ISO 868. Figure 6 shows the graphical dependence of hardness on the percentage ratio of regranulate.

$$HS = 3.93 \cdot 10^{-6} \cdot x^3 - 6.897 \cdot 10^{-4} \cdot x^2 + 0.0276 \cdot x + 67.5 \tag{8}$$

where HS—hardness Shore D, x—percentage of recycled granulate.

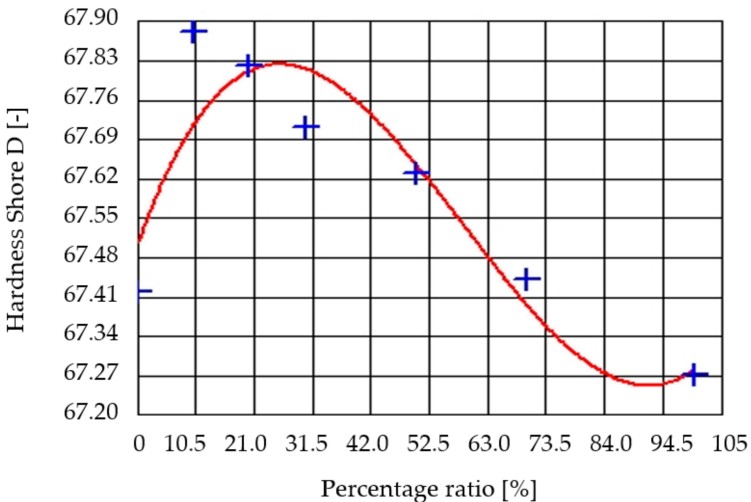

**Figure 6.** Dependence of measured and calculated values of hardness Shore D on volume of recyclate for Mosten GB 005.

The correlation index of measured (blue points) and calculated (red curve) values is 92.39%, dispersion is 0.0438, and standard deviation is 0.209. Measured and average values of Shore hardness are listed in Table 7.

**Table 7.** Measured values of hardness for Mosten GB 005.

| Sample | Number of Material Batch—Percentage Ratio of Recycled Material | | | | | | |
|---|---|---|---|---|---|---|---|
| | **1—0%** | **2—10%** | **3—20%** | **4—30%** | **5—50%** | **6—70%** | **7—100%** |
| 1 | 67.50 | 68.10 | 68.40 | 67.40 | 67.60 | 67.30 | 67.10 |
| 2 | 67.20 | 68.30 | 68.60 | 67.40 | 67.60 | 67.10 | 67.30 |
| 3 | 67.00 | 68.10 | 68.50 | 67.40 | 67.60 | 67.40 | 65.20 |
| 4 | 67.30 | 68.40 | 67.70 | 68.40 | 67.80 | 67.60 | 67.00 |
| 5 | 67.40 | 68.30 | 67.70 | 67.60 | 67.30 | 67.70 | 66.60 |
| 6 | 67.30 | 68.00 | 68.10 | 67.90 | 67.50 | 66.90 | 67.90 |
| 7 | 67.80 | 67.30 | 68.30 | 68.00 | 67.80 | 67.20 | 67.80 |
| 8 | 67.60 | 67.30 | 67.90 | 67.20 | 67.80 | 67.70 | 67.60 |
| 9 | 67.40 | 67.80 | 66.80 | 67.80 | 67.60 | 67.70 | 67.90 |
| 10 | 67.70 | 67.20 | 66.20 | 68.00 | 67.70 | 67.80 | 68.30 |
| **Average Shore D** | **67.42** | **67.88** | **67.82** | **67.71** | **67.63** | **67.44** | **67.27** |

Shore D hardness of the Mosten GB 005 material batch no. 1 reached an average value of 67.42. At the ratio of 10% regranulate in the base material (batch no. 2) Shore D hardness increased to 67.88. For material batch nos. 3–7, an increase in the ratio of the regranulate led to a decrease in the average value of hardness, from 67.82 for material with 20% regranulate (batch no. 3) to 67.27 for 100% regranulate. The results also confirm the conclusions of Tanimoto and Nagakura [26] and Manas et al. [27].

For the evaluation of impact strength, we used impact hammer CEAST Resil 5.5 and software WINMFT. The test was performed in accordance with standard EN ISO 179-1: 2010.

Testing methods were as follows:

- ISO 179-1/1eU—impact strength
- ISO 179-1/1eA—notch toughness

Experimental conditions (ISO 179-1/1eU) were as follows:

- Impact velocity—2.9 m/s ± 10%
- Nominal energy of pendulum—5 J

- Temperature of test sample—23 °C
- Type of fracture—C: full fracture

Table 8 shows measured values of impact strength for a temperature of 23 °C and nominal energy of pendulum 5 J.

**Table 8.** Measured values of impact strength for Mosten GB 005.

| Sample | Number of Material Batch—Percentage Ratio of Recycled Material $a_{cU}$ [kJ/m$^2$] | | | | | | |
|---|---|---|---|---|---|---|---|
| | 1—0% | 2—10% | 3—20% | 4—30% | 5—50% | 6—70% | 7—100% |
| 1–10 | WITHOUT FINAL FRACTURE | | | | | | |

### 3.4. Evaluation of the Impact Strength Test

Based on the test results of the impact strength test, it can be stated that under the current conditions of test material for batch nos. 1–7 at 23 °C, no fracture occurs.

Figure 7 shows impact strength values at −30 °C for measurements of individual batches with variable ratios of regranulate.

$$IS = 1.704 \cdot 10^{-6} \cdot x^3 - 2.598 \cdot 10^{-4} \cdot x^2 + 16.172 \tag{9}$$

where IS—impact strength, x—percentage of recycled granulate.

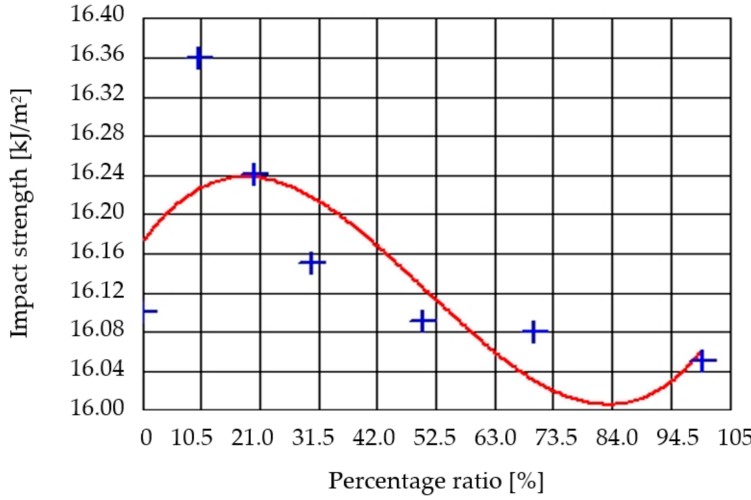

**Figure 7.** Graphical dependence of impact strength on volume of recyclates.

The correlation index of measured (blue points) and calculated (red curve) values is 75.23%, dispersion is 0.01313, and standard deviation is 0.102. Measured and average values of impact strength at −30 °C are listed in Table 9.

Results of the impact strength tests shown in Figure 7; a maximal value of 16.36 kJ.m$^{-2}$ was reached in batch no. 2 with 10% content of regranulate. Further increases of the regranulate ratio in the base material resulted in decreases in values, from 16.36 kJ.m$^{-2}$ for batch no. 2 to a minimum 16.07 kJ.m$^{-2}$ for batch no. 7. The results also confirm the conclusions of the authors Guo et al. [28], Karimipour et al. [29] and Boccardi et al. [30].

Notch toughness was determined according to standard ISO 179-1/1eA under the following conditions:

- Impact velocity—2.9 m/s ± 10%
- Nominal energy of the pendulum—0.5 J
- Temperature of testing sample—23 °C/−30 °C
- Fracture type—C: complete fracture

**Table 9.** Measured values of impact strength for Mosten GB 005 at −30 °C.

| Sample | Number of Material Batch—Percentage Ratio of Recycled Material $a_{cU}$ | | | | | | |
|---|---|---|---|---|---|---|---|
| | 1—0% | 2—10% | 3—20% | 4—30% | 5—50% | 6—70% | 7—100% |
| 1 | 16.025 | 16.300 | 16.825 | 15.850 | 16.175 | 15.875 | 15.725 |
| 2 | 15.975 | 16.150 | 16.075 | 15.725 | 15.775 | 16.450 | 16.350 |
| 3 | 16.225 | 16.300 | 16.175 | 16.075 | 16.175 | 15.725 | 16.075 |
| 4 | 17.150 | 16.525 | 16.250 | 16.600 | 15.550 | 16.250 | 16.175 |
| 5 | 15.950 | 16.200 | 16.075 | 16.350 | 16.075 | 16.600 | 16.250 |
| 6 | 15.725 | 16.150 | 16.175 | 16.000 | 16.175 | 16.450 | 15.875 |
| 7 | 16.550 | 16.525 | 15.825 | 16.075 | 16.050 | 15.975 | 16.350 |
| 8 | 16.075 | 16.300 | 16.250 | 16.600 | 16.175 | 15.975 | 15.975 |
| 9 | 15.600 | 16.450 | 16.550 | 16.175 | 16.550 | 15.875 | 15.875 |
| 10 | 15.725 | 16.650 | 16.175 | 16.075 | 16.175 | 15.625 | 16.075 |
| Average [kJ/m$^2$] | 16.10 | 16.36 | 16.24 | 16.15 | 16.09 | 16.08 | 16.07 |
| Fracture type | C | C | C | C | C | C | C |

Values of notch toughness at temperature 23 °C are shown in Figure 8 as dependence on the percentage ratio of recycled material.

$$NT = -1.36 \cdot 10^{-6} \cdot x^3 + 1.28 \cdot 10^{-4} \cdot x^2 + 3.85 \tag{10}$$

where NT—notch toughness, x—percentage of recycled granulate.

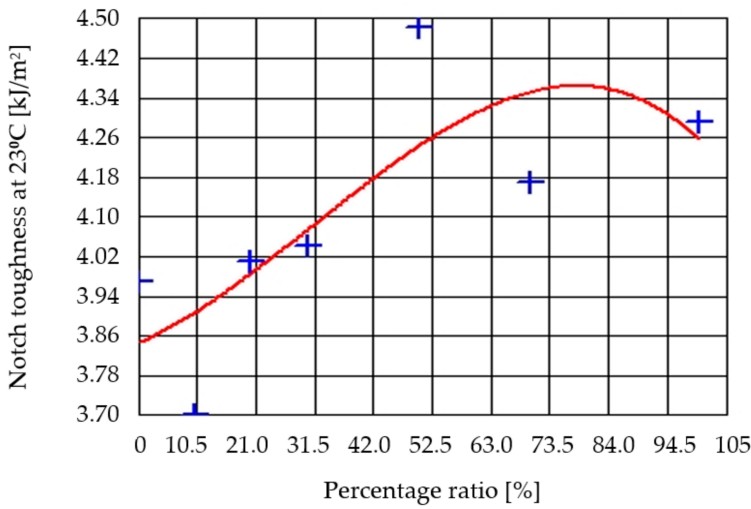

**Figure 8.** Graphical dependence of notch toughness at 23 °C for various percentage ratios of recycled granulate.

Correlation index of measured (blue points) and calculated (red curve) values is 77.03%, dispersion is 0.0533, and standard deviation is 0.2311.

The results of notch toughness as dependence on the percentage ratio of recyclate are shown in Table 10 for a temperature of 23 °C.

The graph in Figure 8 describes the impact of recyclate on notch toughness and values are in the range from 3.70 to 4.48 kJ.m$^{-2}$. The minimum value was recorded in batch no. 2 (10% recyclate): 3.70 kJ.m$^{-2}$. Material batch nos. 3 and 4 had approximately the same value of the notch toughness in comparison with material batch no. 1, at 4.0 kJ.m$^{-2}$. At 50% strength relative to the base granulate material, impact toughness ($a_{cN}$) reached a maximum value of 4.48 kJ.m$^{-2}$. Materials batch nos. 6 and 7 are characterized by a decrease of the maximum values, where the value for 70% recyclate content was 4.17 kJ.m$^{-2}$, and

at 100% it was 4.28 kJ.m$^{-2}$. Values of notch toughness at $-30$ °C are shown in Figure 9 as dependence on the percentage ratio of recycled material.

$$NTC = 0.0031 \cdot x^2 + 1.83293 \tag{11}$$

where NTC—notch toughness at $-30$ °C, x—percentage of recycled granulate.

**Table 10.** Measured values of notch toughness for Mosten GB 005 at 23 °C.

| Sample | Number of Material Batch—Percentage Ratio of Recycled Material a$_{cN}$ | | | | | | |
|---|---|---|---|---|---|---|---|
| | 1—0% | 2—10% | 3—20% | 4—30% | 5—50% | 6—70% | 7—100% |
| 1 | 3.969 | 4.094 | 4.219 | 4.094 | 4.563 | 4.313 | 4.406 |
| 2 | 3.969 | 3.500 | 4.063 | 4.063 | 4.313 | 4.031 | 4.188 |
| 3 | 3.719 | 3.281 | 4.344 | 4.031 | 4.469 | 4.156 | 4.281 |
| 4 | 4.125 | 4.156 | 4.063 | 4.219 | 4.563 | 4.094 | 4.281 |
| 5 | 3.969 | 3.094 | 3.219 | 3.438 | 4.688 | 4.375 | 4.250 |
| 6 | 4.094 | 3.750 | 3.844 | 4.500 | 5.281 | 4.344 | 4.031 |
| 7 | 3.969 | 3.813 | 4.000 | 3.594 | 4.250 | 4.125 | 4.375 |
| 8 | 4.188 | 3.813 | 4.094 | 4.188 | 4.313 | 3.938 | 4.281 |
| 9 | 3.875 | 3.719 | 4.063 | 4.094 | 4.063 | 3.906 | 4.500 |
| 10 | 3.781 | 3.813 | 4.219 | 4.219 | 4.250 | 4.438 | 4.188 |
| Average [kJ/m$^2$] | 3.97 | 3.70 | 4.01 | 4.04 | 4.48 | 4.17 | 4.28 |
| Fracture type | C | C | C | C | C | C | C |

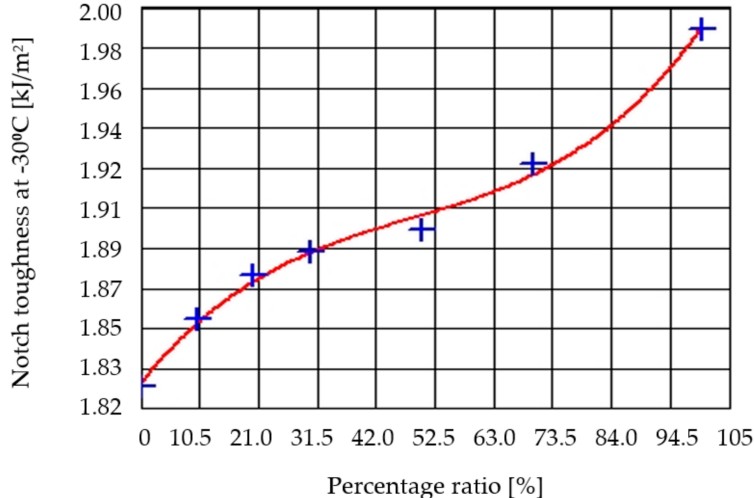

**Figure 9.** Graphical dependence of notch toughness at $-30$ °C for various percentage ratios of recycled granulate.

The correlation index of measured (blue points) and calculated (red curve) values is 99.75%, dispersion is 0.0023, and standard deviation is 0.0477. The results of notch toughness at $-30$ °C for various percentage ratios are shown in Table 11.

The Mosten material was tested at $-30$ °C to establish the impact of recyclate percentage on the values of notch toughness. Increasing the recycle granulate caused an increase of the notch toughness, from 1.83 kJ.m$^{-2}$ for batch no. 1 (0% regrind) to 1.95 kJ.m$^{-2}$ with clear granulates. The results also confirm the conclusions of Lehmann et al. [31].

**Table 11.** Measured values of notch toughness for Mosten GB 005 at −30 °C.

| Sample | Number of Material Batch—Percentage Ratio of RECYCLED material $a_{cN}$ | | | | | | |
|---|---|---|---|---|---|---|---|
| | 1—0% | 2—10% | 3—20% | 4—30% | 5—50% | 6—70% | 7—100% |
| 1 | 1.813 | 1.416 | 1.750 | 1.844 | 2.000 | 1.844 | 2.031 |
| 2 | 1.750 | 1.813 | 2.219 | 1.844 | 1.813 | 2.094 | 1.750 |
| 3 | 1.781 | 1.875 | 2.344 | 1.938 | 2.219 | 2.063 | 2.219 |
| 4 | 2.125 | 1.750 | 1.875 | 1.813 | 2.219 | 2.094 | 1.938 |
| 5 | 1.781 | 2.031 | 1.688 | 1.844 | 1.750 | 2.000 | 2.375 |
| 6 | 1.750 | 2.188 | 1.969 | 1.813 | 1.781 | 1.750 | 1.625 |
| 7 | 1.875 | 2.156 | 1.875 | 1.813 | 1.719 | 1.781 | 1.813 |
| 8 | 1.813 | 2.094 | 1.719 | 1.938 | 2.063 | 1.813 | 1.969 |
| 9 | 1.813 | 1.438 | 1.656 | 2.031 | 1.719 | 2.125 | 2.000 |
| 10 | 1.781 | 1.813 | 1.688 | 2.016 | 1.719 | 1.719 | 1.781 |
| Average [kJ/m²] | 1.83 | 1.86 | 1.88 | 1.89 | 1.90 | 1.93 | 1.95 |
| Fracture type | C | C | C | C | C | C | C |

*3.5. Evaluation of the Flexural Properties*

The flexure modulus was evaluated using tensile machine Hounsfield H10 KT with software Qmat according to standard ISO EN 178:2003.

The evaluated parameters were as follows:

- Flexural modulus $E_f$ (MPa)—the ratio of a differential stress $\sigma_{f2}$–$\sigma_{f1}$ to the value corresponding to the difference deformation $\varepsilon_{f2}$ (0.0005)–$\varepsilon_{f1}$ (0.0025)
- Flexural strength $\sigma_{fM}$ (MPa)—the highest value of the bending stress

The conditions of testing for all experimental samples were as follows:

- Overload—2 N
- Feed rate—2 mm/min
- Distance between supports—64 mm
- Scanning head—500 N

Special treatment of Mosten thermoplastic granules was not necessary before the injection according to the material sheet. Graphical dependence of flexural modulus on the percentage of recycled material is shown in Figure 10.

$$FM = -6.2387 \cdot 10^{-4} \cdot x^3 + 0.106 \cdot x^2 - 3.566 \cdot x + 1357.61 \tag{12}$$

where FM—flexural modulus, x—percentage of recycled granulate.

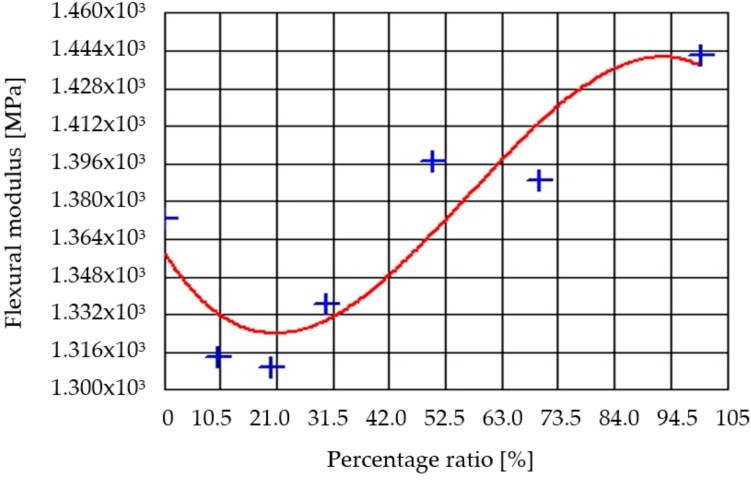

**Figure 10.** Flexural modulus test for Mosten material for various percentages of recycled granulate.

Correlation index of measured (blue points) and calculated (red curve) values is 91.25%, dispersion is 0.67, and standard deviation is 44.84. The results of the flexural modulus testing are shown in Table 12.

**Table 12.** Average values of $E_f$ for Mosten GB 005.

| Sample | Number of Material Batch—Percentage Ratio of Recycled Material | | | | | | |
|---|---|---|---|---|---|---|---|
| | 1—0% | 2—10% | 3—20% | 4—30% | 5—50% | 6—70% | 7—100% |
| 1 | 1530.0 | 1305.0 | 1350.0 | 1350.0 | 1410.0 | 1390.0 | 1485.0 |
| 2 | 1417.5 | 1305.0 | 1305.0 | 1327.5 | 1350.0 | 1395.0 | 1417.5 |
| 3 | 1327.5 | 1305.0 | 1282.5 | 1305.0 | 1395.0 | 1382.5 | 1417.5 |
| 4 | 1282.5 | 1350.0 | 1282.5 | 1350.0 | 1412.5 | 1395.0 | 1462.5 |
| 5 | 1305.0 | 1305.0 | 1327.5 | 1350.0 | 1417.5 | 1382.5 | 1417.5 |
| Average [MPa] | 1372.5 | 1314.0 | 1309.5 | 1336.5 | 1397.0 | 1389.0 | 1440.0 |

The average values of the flexural modulus $E_f$ are shown in Figure 10. The values are in the range from 1309.5 MPa for batch no. 3 to 1440.0 MPa for batch no. 7. Batch no. 1 reached an average value of 1372.5 MPa. The average value decreased to 1314.0 MPa for material batch no. 2 and 1309.5 MPa for batch no. 3. Further increases of the percentage ratio of recycled granulate to the base material caused gradual increases in the values of the flexural modulus $E_f$ for batch nos. 4 and 5. A slight decrease occurred in the material with 70% recyclate, to an average value of 1389.0 MPa; 100% recyclate had the highest value of the flexural modulus $E_f$. The following Figure 11 shows the flexural strength of the Mosten material.

$$FS = -1.3999 \cdot 10^{-4} \cdot x^3 + 0.0016 \cdot x^2 + 0.00426 \cdot x + 41.82 \tag{13}$$

where FS—flexural strength, x—percentage of recycled granulate.

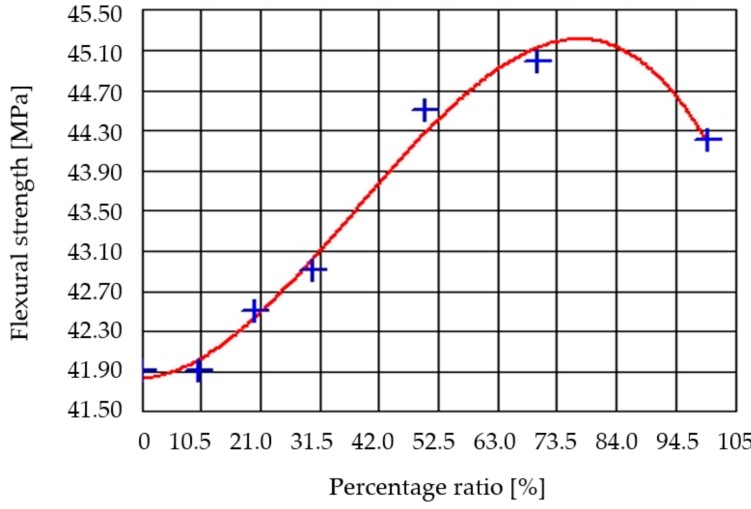

**Figure 11.** Flexural strength of the Mosten material for various percentages of recycled granulate.

The correlation index of measured (blue points) and calculated (red curve) values is 99.79%, dispersion is 1.39, and standard deviation is 1.18. Average values of the experimental results of the flexural strength tests are shown in Table 13.

As shown in Figure 11, the average values of the flexural strength test $\sigma_{fM}$ were in the range from 41.9 MPa for batch nos. 1 and 2 (10% and 20% recyclate) to 44.5 MPa for batch no. 6 (70% recyclate). Increasing the percentage ratio of the recyclate granulate caused an increase the $\sigma_{fM}$ values; batch nos. 3 and 4 had average values of 42.5 MPa and 42.8 MPa, respectively. The most significant increase in the values of flexural $\sigma_{fM}$ was seen in batch no.

5 (50% recyclate), where the average value was around 44.3 MPa. The maximum value was measured at 70% recyclate added into the base material. At 100% recyclate material, the value of $\sigma_{fM}$ decreased by 0.2 MPa compared to the maximum to 44.3 MPa. This behavior was in agreement with data observed by the Espinach et al. [32] and Shibata et al. [33].

**Table 13.** Average values of $\sigma_{fM}$ for Mosten GB 005.

| Sample | Number of Material Batch—Percentage Ratio of Recycled Material | | | | | | |
|---|---|---|---|---|---|---|---|
| | 1—0% | 2—10% | 3—20% | 4—30% | 5—50% | 6—70% | 7—100% |
| 1 | 42.2 | 41.9 | 42.5 | 42.6 | 44.5 | 44.9 | 44.3 |
| 2 | 42.1 | 41.7 | 42.6 | 42.8 | 43.5 | 44.6 | 44.3 |
| 3 | 42.1 | 41.8 | 42.3 | 43.0 | 44.5 | 44.2 | 44.4 |
| 4 | 41.9 | 42.2 | 42.6 | 42.9 | 44.7 | 44.2 | 44.2 |
| 5 | 41.5 | 42.1 | 42.4 | 42.7 | 44.5 | 44.6 | 44.4 |
| **Average [MPa]** | **41.9** | **41.9** | **42.5** | **42.8** | **44.3** | **44.5** | **44.3** |

*3.6. Evaluation of the Thermal Properties*

The DSC (differential scanning calorimetry) method monitors transition temperatures of polymers to detect temperatures of glass transition, crystal melting, and crystallization to fast identification of unknown polymer materials. It is also used to assess the quality of the testing parts by degree of the crystallinity (melting enthalpy auxiliary), to evaluate the cinematic of crystallization, to evaluate copolymers and polymer mixtures, to study the steady state changes, and to evaluate the degradation. Measuring was realized according to standard EN ISO 11357-1:2010.

Experimental research was focused on the evaluation of calorimetric curves for the material Mosten GB 005, with a recycle ratio of 0%, 50%, and 100%, to prescribe the effect of the recycled material on the process of melting, solidification, and crystallization.

Microstructures of the testing material show a significant impact of the recyclate on gains, with an increasing recyclate ratio resulting in finer microstructures (see Figure 12).

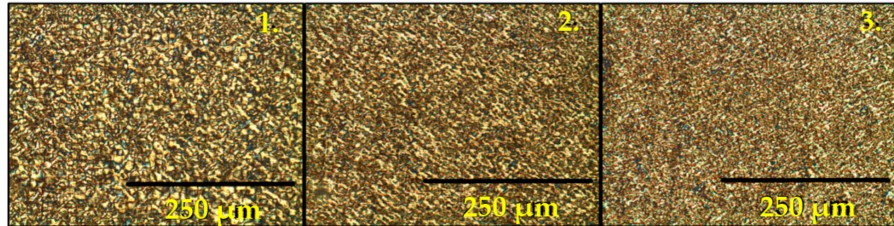

**Figure 12.** Microstructure of Mosten GB 005 (**1.** 0% recyclate, **2.** 50% recyclate, **3.** 100% recyclate).

The graphical dependences shown in Figure 13 represent calorimetric curves for different recyclate ratios (0%, 50%, and 100%). The curves provide information about recyclate impact on calorimetric properties and show that the ratio of recyclate does not significantly affect the calorimetric curves and the material's endothermic and exothermic reaction.

A ratio of recyclate of 0% and 50% represent almost ideal and identical curves of calorimetry, but at 100%, the ratio of recyclate is seen to increase heat flow, although initial and final temperatures for exothermic and endothermic reaction are in the range of 2 °C.

The percentage of the recyclate ratio causes an increase in the initial melting temperature, from 134.676 °C at 0 % to 139.715 °C at 100% ratio of the recyclate. Melting enthalpy was measured at 0 °C, with a value of enthalpy of −122.941 J/g. At 50%, the value was −114.78 J/g, and at 100%, the value was −135.778 J/g. Thus, the smallest amount of the heat supplied to change the physical state was detected at recyclate ratio 50%, with similar results for enthalpy (heat transferred during solidification). This behavior was in agreement with data observed by Majewski et al. [34] and Cheruthazhekatt et al. [35].

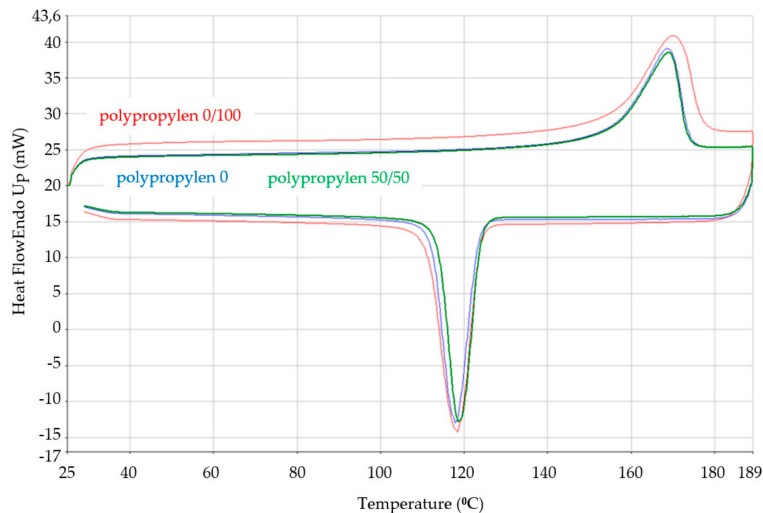

**Figure 13.** Calorimetric curves.

## 4. Conclusions

The article deals with the assessment of the influence of recycled material on the material properties of polypropylene homopolymer. Polypropylene under the trade name Mosten GB 005 was chosen as the test material. Mechanical, rheological, and thermal properties of the material were assessed. The measurement of individual properties was performed according to the relevant international standards.

Table 14 shows a comparison of the results from the material sheet of the original virgin material and the measured results. The results of the test material properties are represented in the table as minimum and maximum values from the measured values of individual batches of material.

**Table 14.** Comparative table of physical and mechanical properties of measured material results.

| Physical Properties | Unit | Nominal Value | Measured Value (Min–Max) |
|---|---|---|---|
| Melt Mass-Flow Rate (MFR) (230 °C/2.16 kg) | (g/10 min) | 5 | 6.98–8.03 |
| **Hardness** | | | |
| Shore Hardness (Shore D) | - | 65 | 67.27–67.88 |
| **Mechanical Properties** | | | |
| Tensile Stress (Yield) | MPa | 34 | 34.4–35.3 |
| Yield | % | 9 | 9.78–9.99 |
| Flexural Modulus | MPa | 1600 | 1309.5–1440.0 |

This work has a significant impact on industrial applications in the field of processing and recycling of plastic materials. The benefit of the research in this article is that conventional recycled material from the production of plastic moldings will be reused in the additive production of plastic products. At the same time, it is important to set technological parameters in the process of production and processing of plastic materials with regard to their further processing. Manufacturers are increasingly trying to use waste material from production in the form of recycled material. They use it as a partial contribution to virgin material in the production of plastic products. Batches of material were selected that represented the weight percentage of recycled material to virgin material. These batches were used to produce test samples. Based on the measured data, evaluations and conclusions were prepared. The use of recyclate also has an important position in the production of various products for outdoor daily use in the form of pure recyclate.

Based on the measured data, the following conclusions were reached:

- The melt volume flow rate was used to analyze the rheological properties. Recycled material has minimal effect on rheological properties. The variance of the measured data was 13.1%.

The assessed mechanical properties (tensile properties, flexural properties, impact properties, hardness) were as follows:

- Yield strength—measured values fluctuated in the range of 2.5%.
- Elongation before fraction—the value increased due to the addition of recycled material by 0.2%.
- Elongation after fraction—the value decreased by 5.6%.
- Tensile strength—measured values fluctuated in the range of 2.5%.
- Hardness—measured values fluctuated in the range of 0.9%.
- Impact strength—at 23 °C there was no puncture of the test specimens. At −30 °C the values fluctuated in the range of 1.8%. For all samples, the full type of puncture C was achieved.
- Notch toughness—at 23 °C the test specimens were completely punctured with type C. Values fluctuated in the range of 17.4%. At −30° C the values fluctuated in the range of 6.2%. The full puncture type C was achieved for all test bodies.
- Flexural strength—measured values fluctuated in the range of 5.8%.

Based on DSC analysis, it was concluded that the addition of recycled material does not affect the thermal properties of the test material. Overall, it was found that the addition of recycled material does not affect or minimally effects the resulting properties of the tested material.

The achieved results were compared with the currently found results of other authors, and several conclusions were reached by discussion. The main contribution of this paper and further scientific and research directions for the authors of the paper are as follows.

The study by Behalek [36] et al. was a pilot for the preparation of this article. The authors Valicek et al. [37] and Ondruska et. al. [38] argued that the behavior of the material during extrusion should be taken into account. This will contribute to better and continuous filling during extrusion in the robot head. This hypothesis was confirmed.

Kang et al. [39] found that the length distribution of glass fibers varies significantly with the combination of recycled composites. Composite reinforced with long glass fibers has been found to restore the degraded mechanical properties encountered in the production of recycled composites. This finding will be examined further. The authors Belviso et al. [40] found that simple mixing of waste and other fillers (without pellet preparation) has an impact on the recycling process. This is the basis for further research into recycled materials.

A research team of authors in the study by Vidakis et al. [41] found that the investigated polymeric material achieved a slight increase in the mechanical properties of the material, both in the tensile and bending modes, while the thermal properties showed an increase in the crystallinity of the polymer during the recycling process. Other authors dealing with polymeric materials were Xiong et al. [42], who examined the properties of polypropylene composites reinforced by non-metals recycled from waste printed circuit boards. Saikrishnan et al. [43] found that the recycling of PP with an additive to a low PE mixture is a potential option for reducing plastic waste but also for recovering the renewable material component from a product that would otherwise be considered waste. Vyncke et. al. [44] found that HDPE has a significant effect on the extent to which rPP can be enhanced by talc. The observed effects of HDPE were not of significant magnitude to correspond to the complete difference in properties between talc-filled rPP and virgin PP. This study helped us to solve another issue of the article.

Many types of commodity plastics can be used in additive technologies, which provide a high-value investment for the use of recycled plastics. However, many of these plastics have different properties than those commonly used in 3D printing. Their testing and research is currently of great benefit for practice in various fields with regard to the environmental aspect. Zander et al. [45] found in their paper that many types of waste

plastic materials can be used in additive technologies. Waste materials can be reused in the production process. Many of the polymers studied have reduced mechanical properties and increased waviness/shrinkage compared to those commonly used materials in 3D printing.

**Author Contributions:** J.D. conceived and designed the experiments; M.P. analyzed and evaluated data, processed the data and wrote the paper; L.B. performed the experiments and measurements; J.S. conceptualization the paper. All authors have read and agreed to the published version of the manuscript.

**Funding:** This research received no external funding.

**Institutional Review Board Statement:** Not applicable.

**Informed Consent Statement:** Not applicable.

**Data Availability Statement:** Not applicable.

**Acknowledgments:** This paper has been elaborated in the framework of the project VEGA no. 1/0026/19.

**Conflicts of Interest:** The authors declare no conflict of interest. The founding sponsors had no role in the design of the study; in the collection, analyses, or interpretation of data; in the writing of the manuscript, and in the decision to publish the results.

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
