# Peer review of "Implementation of a Recycled Polypropylene Homopolymer Material for Use in Additive Manufacturing"

_sustainability, doi:10.3390/su13094990_

Round 1

Reviewer 1 Report

The aim of the article is to assess the impact of adding technological waste to virgin material on the resulting material properties of polypropylene homopolymer. The research is carried out in order to capture the possible adverse effect of the added recyclate on the mentioned material properties. The tested material was selected based on practical experience and its use in the automotive industry and other areas. The main benefit of using recycled material is the positive impact on the environment and renewable energy sources. The methodology used in the article is applicable to all types of thermoplastic materials. Through it, were monitored the change of important material properties due to the addition of recycled material. The article is interesting and suitable for publication in this journal, but several adjustments need to be made for its better quality.

I recommend making these changes in the article:

  • In the references of the article, I recommend partially deepening the scope of references by adding important publications in the field of research.
  • In the graphs, I recommend changing the values defined by comma to a decimal point, as stated in the entire text of the article.
  • In the performed tests on page 4 the measurement of thermal properties is not mentioned. It should be mentioned there.
  • It is necessary to specify the use of recycled plastic material in additive technologies as mentioned in the title of the article.

Author Response

Reviewer 1

Dear reviewer,

thank you very much for your comments on the article. In this annex I attach the changes made to the article. I respond to your comments below:

Comments and Suggestions for Authors

The aim of the article is to assess the impact of adding technological waste to virgin material on the resulting material properties of polypropylene homopolymer. The research is carried out in order to capture the possible adverse effect of the added recyclate on the mentioned material properties. The tested material was selected based on practical experience and its use in the automotive industry and other areas. The main benefit of using recycled material is the positive impact on the environment and renewable energy sources. The methodology used in the article is applicable to all types of thermoplastic materials. Through it, were monitored the change of important material properties due to the addition of recycled material. The article is interesting and suitable for publication in this journal, but several adjustments need to be made for its better quality.

I recommend making these changes in the article:

  • In the references of the article, I recommend partially deepening the scope of references by adding important publications in the field of research.

The reference list was supplemented by important publications in the field of research.

  • In the graphs, I recommend changing the values defined by comma to a decimal point, as stated in the entire text of the article.

All images throughout the article have been edited.

  • In the performed tests on page 4 the measurement of thermal properties is not mentioned. It should be mentioned there.

The missing mention of the implemented thermal tests was added in the text of the article – line 170.

  • It is necessary to specify the use of recycled plastic material in additive technologies as mentioned in the title of the article.

A description of the tested material intended for additive production with a description of its industrial use was added to the conclusions of the article - see lines 488-499.

Reviewer 2 Report

This article proposes a study to know the mechanical properties of a recycled material for the use in additive manufacturing.

  1. Increase the number of references to contemporary literature.
  2. Avoid: page 183 “The following Figure 3”. Revise all the manuscript.
  3. Another important issue associated with this manuscript submitted to Sustainability is the in-depth discussion. The current version looks more like a good technical report rather than an academic journal paper. The authors would be appreciated if they can give some discussion on the significance of the presented technology to the additive manufacturing.
  4. Create a comparative table of mechanical properties with a conventional material and this material when they are used in additive manufacturing processes. Discuss the improvements.
  5. In the conclusions, discuss the benefits and/or drawbacks (quantitative) of this new material in an additive process respect the conventional material.
  6. The conclusions are general and they should be improved.

Author Response

Reviewer 2

Dear reviewer,

thank you very much for your comments on the article. In this annex I attach the changes made to the article. I respond to your comments below:

Comments and Suggestions for Authors

This article proposes a study to know the mechanical properties of a recycled material for the use in additive manufacturing.

  1. Increase the number of references to contemporary literature.

The reference list was supplemented by important publications in the field of research.

  1. Avoid: page 183 “The following Figure 3”. Revise all the manuscript.

All adjustments have been made throughout the article.

  1. Another important issue associated with this manuscript submitted to Sustainability is the in-depth discussion. The current version looks more like a good technical report rather than an academic journal paper. The authors would be appreciated if they can give some discussion on the significance of the presented technology to the additive manufacturing.

This work has a significant impact on industrial applications in the field of processing and recycling of plastic materials. The benefit of the research in this article is that conventional recycled material from the production of plastic moldings will be reused in the additive production of plastic products. At the same time, it is important to set technological parameters in the process of production and processing of plastic materials with regard to their further processing. Manufacturers are increasingly trying to use waste material from production in the form of recycled material. They use it as a partial contribution to virgin material in the production of plastic products. Batches of material were selected that represented the weight percentage of recycled material to the virgin material. Among these batches were produced test samples. Based on the measured data, evaluations and conclusions were prepared. The use of recyclate also has an important position in the production of various products in outdoor use of daily use in the form of pure recyclate.

This description has been added to the article - see lines 488-499.

  1. Create a comparative table of mechanical properties with a conventional material and this material when they are used in additive manufacturing processes. Discuss the improvements.

Table 1 shows the properties of the virgin material. The properties of the virgin material were compared with the properties that arose after the preparation and testing of individual batches of the material after the addition of recycled material. The graphical dependencies are also shown in Figures 1 to 13.

  1. In the conclusions, discuss the benefits and/or drawbacks (quantitative) of this new material in an additive process respect the conventional material.

Benefits and potentially drawbacks have been added and discussed in the conclusions chapter of the article.

  1. The conclusions are general and they should be improved.

The conclusions were improved by a discussion and compared with the achieved results of other authors from the same field of research and technology – see lines 525-561.

Reviewer 3 Report

Almost the whole paper can be found online from an article published 2016.
https://sciforum.net/paper/view/conference/3396
I conclude that this article has been published 5 years ago and therefore, the publicized results aren't new.
Besides, I think it is self-plagiarism.

Author Response

Dear reviewer.

I comunicated about this with Section Managing editor from MDPI.

The journal has confirmed that my paper was accepted
to be considered for publication through a peer-review process.

So can we continue?

Have a nice day

Round 2

Reviewer 2 Report

This article proposes a study to know the mechanical properties of a recycled material for the use in additive manufacturing.

  1. Correct: lines 184, 222, etc. i.a.,“The following Figure 3”. Revise all the manuscript.
  2. Create a quantitative and comparative table of mechanical properties to discuss the improvements with percentages. Otherwise, the study seems a technical report.
  3. I can not see from the line 525 to the line 561. It seems an introduction.

Author Response

Reviewer 2_Round 2

Dear reviewer,

thank you very much for your comments on the article. In this annex I attach the changes made to the article. I respond to your comments below:

Comments and Suggestions for Authors

This article proposes a study to know the mechanical properties of a recycled material for the use in additive manufacturing.

  1. Correct: lines 184, 222, etc. i.a.,“The following Figure 3”. Revise all the manuscript.

Mentioned lines were revised in all the manuscript.

  1. Create a quantitative and comparative table of mechanical properties to discuss the improvements with percentages. Otherwise, the study seems a technical report.

A table of physical and mechanical properties was added to the article with measured results of individual batches of final material tests for comparison with the original virgin material. Table 14 shows the values from the material sheet of the original virgin material with a comparison of the results obtained by measurement. The stated values represent the measured results of the minimum and maximum values of the material properties of the individual tested batches of material.

Table 14. Comparative table of physical and mechanical properties of measured material results.

Physical Properties

Unit

Nominal Value

Measured Value

(min-max)

Melt Mass-Flow Rate (MFR) (230°C/2.16 kg)

(g/10 min)

5

6.98 – 8.03

Hardness

Shore Hardness (Shore D)

-

65

67.27 – 67.88

Mechanical Properties

Tensile Stress (Yield)

MPa

34

34.4 – 35.3

Yield

%

9

9.78 – 9.99

Flexural Modulus

MPa

1600

1309.5 – 1440.0

  1. I can not see from the line 525 to the line 561. It seems an introduction.

Expressions in lines 525-561 represent a concrete comparison of the achieved results with other authors. The conclusion describes the benefits and potentially drawbacks. The conclusion also presents a summary of the results in comparison with the initial data on the material and its properties in a comparison table and descriptively in the individual points of the measurement results. The introduction presents a current description of the implemented experiments with references of important publications in the field of research for the elaboration of an overview of the solution on this problem in the world.

Round 3

Reviewer 2 Report

-